# Bearing Fault Diagnosis Using a Hybrid Fuzzy V-Structure Fault Estimator Scheme

**DOI:** 10.3390/s23021021

**Published:** 2023-01-16

**Authors:** Farzin Piltan, Jong-Myon Kim

**Affiliations:** 1Department of Electrical, Electronics and Computer Engineering, University of Ulsan, Ulsan 44610, Republic of Korea; 2PD Technologies Corporation, Ulsan 44610, Republic of Korea

**Keywords:** bearing, vibration data, V-structure technique, autoregressive technique, fuzzy algorithm, support vector method, Laguerre filter, fault classification, crack size identification

## Abstract

Bearings are critical components of motors. However, they can cause several issues. Proper and timely detection of faults in the bearings can play a decisive role in reducing damage to the entire system, thereby reducing economic losses. In this study, a hybrid fuzzy V-structure fuzzy fault estimator was used for fault diagnosis and crack size identification in the bearing using vibration signals. The estimator was designed based on the combination of a fuzzy algorithm and a V-structure approach to reduce the oscillation and improve the unknown condition’s estimation and prediction in using the V-structure method. The V-structure surface is developed by the proposed fuzzy algorithm, which reduces the vibrations and improves the stability. In addition, the parallel fuzzy method is used to improve the robustness and stability of the V-structure algorithm. For data modeling, the proposed combination of an external autoregression error, a Laguerre filter, and a support vector regression algorithm was employed. Finally, the support vector machine algorithm was used for data classification and crack size detection. The effectiveness of the proposed approach was evaluated by leveraging the vibration signals provided in the Case Western Reserve University bearing dataset. The dataset consists of four conditions: normal, ball failure, inner fault, and outer fault. The results showed that the average accuracy of fault classification and crack size identification using the hybrid fuzzy V-structure fuzzy fault estimation algorithm was 98.75% and 98%, respectively.

## 1. Introduction

Bearings are system parts that generally reduce friction. The study of these components has significantly advanced in recent years owing to their many applications. Bearings are used in various industries, such as oil refining, gas transportation, and power generation. Early bearing anomaly detection can improve a system’s performance and increase operation safety. Generally, three types of failures in bearings have been analyzed: ball, inner, and outer faults. Diverse types of sensors can be selected to exploit the normal and abnormal data from bearings, such as vibration sensors, acoustic emission (AE) sensors, current sensors, and voltage sensors. In recent years, vibration and AE sensors have been most generally applicable in industries [1,2].

For bearing fault diagnosis, various methods have been used. They can be categorized into four basic groups: model-based, signal processing-based, artificial-intelligence (AI)-based, and hybrid approaches [3]. The model-based technique is designed based on the mathematical modeling of systems/signals. Most of the model-based techniques are robust and stable; however, they are strongly dependent on the system model. The signal processing-based techniques are algorithms that are evaluated based on the signal processing methods in the time domain, frequency domain, and time-frequency domain. Signal processing-based algorithms have limitations in improving robustness in uncertain conditions. The AI-based techniques are developed based on machine learning, deep learning, and fuzzy algorithms. Meanwhile, AI-based approaches that are used in numerous applications for fault diagnosis require a huge dataset. To improve the performances of model-based, signal processing-based, and AI-based techniques, hybrid-based algorithms have been recommended for various applications. The hybrid algorithms are designed using a combination of the above algorithms [4,5,6].

The application of AI for fault detection and isolation is discussed in refs. [7,8,9], and the application of fuzzy logic algorithms, neural networks, machine learning, and deep learning algorithms for fault diagnosis is described in refs. [10,11,12,13,14]. However, robustness and optimization of the rule table are two important challenges for fuzzy-based algorithms. To improve the performance of the fuzzy-based algorithm, neural network algorithms and neuro-fuzzy approaches are suggested in refs. [11,12]. In addition, the application of machine learning and deep learning algorithms for fault diagnosis is addressed in refs. [13,14]. For example, Prosvirin et al. [15] designed a deep learning-based observation technique for blade-rub impact fault identification. Moreover, the combination of an autoencoder technique and a convolutional neural network for bearing fault diagnosis under different operating conditions has been explained in ref. [16]. On the other hand, the application of signal processing-based algorithms for fault detection and isolation is discussed in refs. [17,18,19]. Furthermore, to design bearing fault diagnosis techniques, various approaches have been carried out that can be used in three main domains: the time domain, the frequency domain, and the time–frequency domain. Time- and frequency-domain analyses have the challenge of high feature dimensions. Time–frequency domain analysis, such as the short-time Fourier transform [17], Wigner Ville distribution [18], and wavelet packet transform [19], have been recommended to solve problems associated with the nonstationary and nonlinear nature of the bearings’ signals.

Numerous algorithms can be introduced as model-based approaches that are categorized into three main groups: data estimation algorithm techniques, output observation approaches, and data identification methods. The application of the Kalman filter (KF) observer for fault diagnosis of the lithium-ion battery is presented in ref. [20]. Based on that article, the adaptive technique has been recommended to improve the robustness of the KF in uncertain conditions. Moreover, the application of an unknown input observer for fault diagnosis is presented in ref. [21]. However, although that technique is robust, its complexity is a challenge. In addition, the fault detection filter observer for fault diagnosis in industrial applications has been analyzed in ref. [22]. This algorithm is used to detect unknown conditions in the state and output them using different types of filtering techniques. Complexity and robustness are two important challenges in refs. [21,22]. The observation technique has been introduced to improve the performance of KF techniques.

Furthermore, using the interval observer for a linear parameter system is discussed in ref. [23]. With this technique, faults can be detected by parameter deviation approximation. The limitation of this technique is that the unknown conditions and faults must be bounded [23]. The observation approach can be divided into two important groups: linear observers [24,25] and nonlinear observers [26,27,28,29,30,31,32]. The application of linear observers is presented in refs. [24,25]. The robustness and reliability are the main drawbacks of linear observers. To solve the challenge of linear observers, nonlinear observation techniques, such as sliding mode (variable structure) observers [26,27,28,29,30] and feedback linearization observers [31,32], have been suggested. The sliding mode observer is a type of variable structure (V-structure) observer that can be used for fault detection and isolation and is presented in ref. [26]. Despite its reliability and robustness, this technique suffers from the chattering phenomenon and system dependency. To resolve these challenges in the sliding mode observer/V-structure, higher-order techniques [27], the super-twisting algorithm [28], the smooth-sliding mode observer [29], and the combined AI and sliding-mode observer [30] have been recommended. The main challenge of higher-order techniques [27] and the super-twisting algorithm [28] is complexity. In addition, the smooth-sliding mode observer [29] and the combined AI and sliding-mode observer [30] have been recommended to solve the complexity; however, robustness is the main problem of these techniques. Moreover, the application of a feedback linearization observer for anomaly detection and isolation is presented in ref [31]. However, a strong dependence on the system model and robustness are the greatest limitations of this approach. To reduce the challenge of modeling dependency, the combination of AI and a feedback linearization observer has been presented in ref. [32].

Although model-based approaches are stable and reliable, a key challenge of these approaches is data and system modeling. Two important algorithms can be used for data modeling: mathematical-based approaches and data identification algorithms [29]. Mathematical-based data modeling is accurate and reliable. Nevertheless, this technique is notably complex, especially for highly nonlinear systems [32]. In data identification techniques, the data can be modeled using machine learning algorithms and regression techniques. The autoregressive technique and the autoregressive approach with external input are essential approaches for data identification using regression methods. To improve the reliability and robustness of regression algorithms, the combination of regression methods and filters, such as autoregression with an external input Laguerre filter, has been recommended in ref. [33]. Another method to improve the performance of data modeling using regression algorithms is the combination of regression techniques and AI algorithms, such as nonlinear autoregression with external input, which is suggested in ref. [34].

In this study, a hybrid technique was leveraged for fault diagnosis in the bearing. This hybrid approach is based on the combination of hybrid data modeling, hybrid data estimation, and data classification. The main advantages of the proposed technique compared with the existing methods are robustness, a low rate of high-frequency oscillation, and reliability. The V-structure technique is a robust algorithm, but this technique suffers from high-frequency oscillation. To reduce the challenge of chattering, the V-structure surface is designed by the proposed Proportional-Integral-Derivative (PID) fuzzy with minimum fuzzy rules and complexity. In addition, to reduce the challenge of uncertainties in the V-structure technique, the fuzzy technique is suggested. In this part, the fuzzy algorithm is selected to increase the flexibility of the proposed method. For hybrid-based bearing data modeling, the combination of the autoregressive error method, a Laguerre filter, and a support vector regression (SVR) were employed. First, the autoregressive technique was used for bearing data modeling. To reduce the data modeling error rate and decrease the order of the autoregression technique, the external error input was applied to the autoregression technique. Moreover, the Laguerre filter was used to improve the robustness of the bearing data modeling. Finally, to improve the nonlinear behavior of the bearing data modeling, the SVR technique was applied to the method of autoregression combined with the error input of the Laguerre approach. For hybrid-based bearing data estimation, the combined V-structure observation algorithm, fuzzy high frequency reduction, and fuzzy fault estimation improvement were employed. First, the robust V-structure observer was used for bearing data estimation. To increase the accuracy and reduce high-frequency oscillation, the PID fuzzy method was used with the V-structure observer. Lastly, the fault estimator was improved by applying the fuzzy fault estimation technique to the previous part of the bearing data estimation. Furthermore, the support vector machine (SVM) was leveraged for fault detection and isolation. The main contributions of the proposed hybrid fuzzy V-structure fuzzy fault estimation technique are outlined below.

A hybrid bearing vibration data modeling method using a robust autoregressive-Laguerre SVM regression algorithm was created.A robust hybrid vibration data estimation approach using the fuzzy V-structure fuzzy fault estimator was designed.SVM was applied to the proposed robust fuzzy V-structure fuzzy fault estimator approach for data classification and data crack size identification.

The remainder of this paper is organized as follows: the experimental Case Western Reserve University bearing dataset is explained in Section 2. The design of the proposed data modeling, data estimation, and data classification is described in Section 3. The results and discussion for bearing fault detection and classification are provided in Section 4. Finally, in Section 5, conclusions and future work are discussed.

## 2. Experimental Dataset

The Case Western Reserve University Bearing Dataset (CWRUBD) was used to experimentally test the combination of the proposed fuzzy V-structure fuzzy fault estimator and SVM, which is illustrated in Figure 1 [31].

This experimental data acquisition was used to extract the bearing data in normal and abnormal conditions. A 2 hp Reliance electric induction motor is used to supply the bearing’s energy. These bearings are used to support the motor shaft. In addition, the dynamometer and electric control system are used to apply the torque to the shaft. To collect the acceleration data, a vibration sensor (6205-2RS JEM SKF) was employed. This vibration sensor collected data under four conditions: bearing in a normal condition (N), bearing in a ball faulty condition (B), bearing in an inner faulty condition (I), and bearing in an outer faulty condition (O). The 16-channel data equation (DA) recorder is used for data collection, and it was processed in a MATLAB environment. The sampling rate for the bearing data collection was 12 kHz samples per second for four different torque loads between 0 horsepower (hp) and 3 hp, with motor speeds of 1720 to 1797 rpm. Moreover, single-point faults with three different crack sizes of 0.007 in, 0.014 in, and 0.021 in were collected in CWRUBD [31]. Furthermore, the depth of the fault was 0.011 inches for all the bearing faults. Table 1 provides a summary of the bearing dataset.

Figure 2 presents the original raw data for the bearing in the normal and three abnormal conditions.

## 3. Proposed Scheme

The proposed algorithm for bearing anomaly classification and crack size detection is illustrated in Figure 3. As shown in the figure, the proposed bearing fault diagnosis method has three main parts: (1) data modeling, (2) data estimation, and (3) data classification and size identification.

For data modeling, the combination of the autoregression external error, Laguerre filter, and SVR algorithm is proposed. The autoregression method is a linear regression algorithm for data modeling. To decrease the data modeling error rate and reduce the order of the autoregression technique, the external error input is applied to the autoregression technique. To improve the robustness of data modeling, the Laguerre filter is applied to the autoregression external error (AE) to utilize the AE with the Laguerre filter (AEL). After improving the robustness of data modeling to improve the power of nonlinear condition modeling, the SVR technique is applied to the AEL method, and AE is utilized with the Laguerre filter in parallel with the SVR (AEL) technique.

For data estimation, the combination of the V-structure estimator, fuzzy high-frequency reduction, and fuzzy fault estimation improvement is proposed. The V-structure (VS) estimator is a robust, reliable, and nonlinear estimator. This estimator has three parts: (1) a nonlinear section, (2) a linear-based part, and (3) a fault (unknown information) estimation part. The nonlinear section is extracted from data modeling. To improve the performance of the linear-based part and reduce the high-frequency oscillation, the proportional-integral-derivative (PID) fuzzy method is applied to the VS estimator, and the FVS approach is utilized. Furthermore, to modify the fault estimation part, the fuzzy algorithm is recommended in parallel with FVS and the application of the FVSF technique.

The data classification and size identification section have two parts. After determining the residual data using the difference between the bearing raw data and estimated data using the proposed algorithm, the root means square (RMS) data are calculated. Furthermore, the SVM algorithm is used for data classification and crack size detection. As shown in Figure 3, we use two levels of SVM: the first is for data classification, and the second is to determine the crack sizes.

### 3.1. Data Modeling

The first step in designing the proposed scheme for bearing fault diagnosis is data modeling. For data modeling, the combination of the autoregression external error, a Laguerre filter, and the SVR algorithm is proposed. The autoregression method is a linear regression algorithm for modeling the data under normal conditions. This approach is linear and intended to reduce the error of data modeling and the order of the autoregression technique; the external error input is applied to the autoregression technique. Although the combination of the autoregressive and feedback error inputs improves the performance of data modeling, the Laguerre filter is applied to the autoregression external error (AE) and the AE is utilized with the Laguerre filter (AEL) to improve the robustness of data modeling. After improving the robustness of data modeling to improve the power of nonlinear condition modeling, the combination of the SVR technique and the AEL approach, AELS, is suggested. The basic concept for data modeling in this work is a regression algorithm. First, the autoregression (A) technique is defined using the following definition:(1)$$\{\begin{array}{l} \gamma_{a}\left(k+1\right)=\left[\alpha_{\gamma }\gamma_{a}\left(k\right)+\alpha_{i}I_{i}\left(k\right)\right]+\emptyset_{a}\left(k\right)\\ \beta_{a}\left(k\right)={\left(\alpha_{o}\right)}^{T}\gamma_{a}\left(k\right) \end{array}$$

Here, $\gamma_{a}\left(k\right)$ is the state-space of the bearing vibration data modeling based on the autoregression technique, $I_{i}\left(k\right)$ is the bearing vibration data, $\beta_{a}\left(k\right)$ is the output of the bearing vibration data modeling based on the autoregression technique, and $\left(\alpha_{\gamma },\alpha_{i},\alpha_{o}\right)$ are the coefficients for tuning the autoregressive technique. Moreover, $\emptyset_{a}\left(k\right)$ is the unknown (uncertain) condition of the bearing vibration data modeling based on the autoregression technique, and it is introduced by the following definition:(2)$$\emptyset_{a}\left(k\right)=I_{i}\left(k\right)-\beta_{ae}\left(k\right)$$

To modify the autoregression technique, AE is used in the next part.
(3)$$\{\begin{array}{l} \gamma_{ae}\left(k+1\right)=\left[\alpha_{\gamma }\gamma_{ae}\left(k\right)+\alpha_{i}I_{i}\left(k\right)\right]+\emptyset_{ae}\left(k\right)+\alpha_{e}e_{ae}\left(k\right)\\ \beta_{ae}\left(k\right)={\left(\alpha_{o}\right)}^{T}\gamma_{ae}\left(k\right) \end{array}$$
where $\gamma_{ae}\left(k\right)$ is the state-space of the bearing vibration data modeling based on the AE technique, $e_{ae}\left(k\right)$ is the error of the bearing vibration data modeling based on the AE technique, $\beta_{ae}\left(k\right)$ is the output of the bearing vibration data modeling based on that technique, and $\emptyset_{ae}\left(k\right)$ is the unknown (uncertain) condition of the bearing vibration data modeling based on that technique. In addition, $\left(\alpha_{e}\right)$ is the coefficient for tuning the technique. Here, $e_{ae}\left(k\right)$ and $\emptyset_{ae}\left(k\right)$ are introduced by the following equation:(4)$$\{\begin{array}{l} \emptyset_{ae}\left(k\right)=I_{i}\left(k\right)-\beta_{ae}\left(k\right)\\ e_{ae}\left(k\right)=\beta_{ae}\left(k\right)-\beta_{ae}\left(k-1\right) \end{array}$$

To improve the robustness of the bearing vibration data modeling in the AE technique, the AE with the Laguerre filter (AEL) technique is introduced in the next stage.
(5)$$\{\begin{array}{l} \gamma_{ael}\left(k+1\right)=\left[\alpha_{\gamma }\gamma_{ael}\left(k\right)+\alpha_{i}I_{i}\left(k\right)+\alpha_{\beta }\beta_{ael}\left(k\right)\right]+\emptyset_{ael}\left(k\right)+\alpha_{e}e_{ael}\left(k\right)\\ \beta_{ael}\left(k\right)={\left(\alpha_{o}\right)}^{T}\gamma_{ael}\left(k\right) \end{array}$$

Here, $\gamma_{ael}\left(k\right)$ is the state-space of the bearing vibration data modeling based on the AE with the Laguerre filter technique, $e_{ael}\left(k\right)$ is the error of the bearing vibration data modeling based on the AE with the Laguerre filter technique, $\beta_{ael}\left(k\right)$ is the output of the bearing vibration data modeling based on the AE with the Laguerre filter technique, $\emptyset_{ael}\left(k\right)$ is the unknown (uncertain) condition of the bearing vibration data modeling based on the AE with the Laguerre filter technique, and $\left(\alpha_{\beta }\right)$ is the coefficient for tuning the AE with the Laguerre filter technique. Additionally, $e_{ael}\left(k\right)$ and $\emptyset_{ael}\left(k\right)$ are introduced by the following equation:(6)$$\{\begin{array}{l} \emptyset_{ael}\left(k\right)=I_{i}\left(k\right)-\beta_{ael}\left(k\right)\\ e_{ael}\left(k\right)=\beta_{ael}\left(k\right)-\beta_{ael}\left(k-1\right) \end{array}$$

Moreover, to improve the nonlinear conditions for the bearing vibration data modeling in the AE with the Laguerre filter technique, the combination of the AE with the Laguerre filter and SVR technique (AELS) is presented in the next stage.
(7)$$\{\begin{array}{l} \gamma_{aels}\left(k+1\right)=\left[\alpha_{\gamma }\gamma_{aels}\left(k\right)+\alpha_{i}I_{i}\left(k\right)+\alpha_{\beta }\beta_{aels}\left(k\right)\right]+\emptyset_{ael}\left(k\right)+\alpha_{e}e_{aels}\left(k\right)+\alpha_{s}\beta_{SVR}\left(k\right)\\ \beta_{aels}\left(k\right)={\left(\alpha_{o}\right)}^{T}\gamma_{aels}\left(k\right) \end{array}$$
where $\gamma_{aels}\left(k\right)$ is the state-space of the bearing vibration data modeling based on the combination of the AE with the Laguerre filter and SVR technique, $e_{aels}\left(k\right)$ is the error of the bearing’s vibration data modeling based on the combination of the AE with the Laguerre filter and SVR technique, $\beta_{aels}\left(k\right)$ is the output of the bearing vibration data modeling based on the combination of AE with the Laguerre filter and SVR technique, $\beta_{SVR}\left(k\right)$ is the output of the error vibration data modeling based on the SVR algorithm, $\emptyset_{aels}\left(k\right)$ is the unknown (uncertain) condition of the bearing’s vibration data modeling based on the combination of AE with the Laguerre filter and SVR technique, and $\left(\alpha_{s}\right)$ is the coefficient for tuning the combination of AE with the Laguerre filter and SVR technique. In addition, $e_{aels}\left(k\right)$ and $\emptyset_{aels}\left(k\right)$ are introduced by the following equation:(8)$$\{\begin{array}{l} \emptyset_{aels}\left(k\right)=I_{i}\left(k\right)-\beta_{aels}\left(k\right)\\ e_{aels}\left(k\right)=\beta_{aels}\left(k\right)-\beta_{aels}\left(k-1\right) \end{array}$$

To compensate for the nonlinearity of the bearing’s data, the SVR technique is suggested and introduced using the following definition:(9)$$\beta_{SVR}\left(k\right)=\sum_{j}\left(\sigma_{j}{}^{+}-\sigma_{j}{}^{-}\right)∁\left(\varepsilon_{i},\varepsilon \right)+\omega$$

Here, $\left(\sigma_{j}{}^{+}-\sigma_{j}{}^{-}\right)$ is the constant for the Lagrange function, $∁\left(\varepsilon_{i},\varepsilon \right)$ is a nonlinear kernel function, and ω is the bias of the function. The nonlinear Gaussian function is suggested for the kernel in this work, and it can be introduced using the following definition:(10)$$∁\left(\varepsilon_{i},\varepsilon \right)=e^{\left(-\frac{1}{2\partial^{2}}\|\sigma_{j}{}^{+}-\sigma {\|}^{2}\right)}$$
where ∂ is the variance. Moreover, the bias of the function can be introduced using the following definition:(11)$$\omega =\frac{1}{\left|S\right|}\sum_{s\in S}\left[\beta_{S}-\sum_{j\in S}\left(\sigma_{j}{}^{+}-\sigma_{j}{}^{-}\right)\times ∁\left(\varepsilon_{i},\varepsilon \right)-(\mu \times sgn\left(\sigma_{j}{}^{+}-\sigma_{j}{}^{-}\right)\right)]$$

Here, $\beta_{S}$ is the support vector signal, μ is the accepted boundary for the support vector for signal compensation, and S is an approximation support vector. Furthermore, the approximation support vector can be introduced in the following range:(12)$$S=\left\{j|0<\sigma_{j}{}^{+}-\sigma_{j}{}^{-}<\Delta \right\}$$

Here, Δ is the constant that is used to calculate the range of the approximation support vector. Regarding Figure 3, the vibration’s bearing data is modeled using the combination of AE with the Laguerre filter and SVR technique (AELS). Table 2 outlines the steps for designing the proposed data modeling using the combination of AE with the Laguerre filter and SVR technique, ALES. In the next section, the hybrid robust estimator is introduced to estimate the bearing’s data under unknown conditions.

### 3.2. Data Estimation

After modeling the normal data using the AELS approach, for data estimation (observation), the combination of the V-structure estimator, fuzzy V-structure surface for high-frequency reduction, and fuzzy approach for fault estimation improvement is proposed. The V-structure (VS) estimator is a robust, reliable, and nonlinear estimator. This estimator has three parts: (1) nonlinear section, (2) linear-based part, and (3) fault (unknown information) estimation part. The nonlinear section is extracted from the data modeling explained in the previous part. The linear part can be designed based on the linear proportional-integral-derivative (PID) controller, proportional-integral (PI) controller, and proportional-derivative (PD) controller. In this work, the PID technique is suggested. The challenge of a linear-based V-structure surface is high frequency oscillation. To improve the performance of the linear-based part and reduce the high-frequency oscillation, the proposed proportional-integral-derivative (PID) fuzzy method with a minimum rule base is applied to the VS estimator, and the FVS approach is utilized. Furthermore, to modify the fault estimation part, the fuzzy algorithm is used in parallel with the FVS technique to design the FVSF approach. Based on Figure 3, this algorithm has three parts: (1) nonlinear functions that must be extracted from vibration data using the proposed modeling algorithm, (2) linear-based part to improve and tune the V-structure approach, and (3) fault prediction, which is used to reduce the error of signal estimation and increase the ability to detect vibration data in normal and abnormal conditions. The classical V-structure algorithm is defined as follows:(13)$$\{\begin{array}{l} \gamma_{vs}\left(k+1\right)=\left[\alpha_{\gamma }\gamma_{aels}\left(k\right)+\alpha_{i}I_{i}\left(k\right)+\alpha_{\beta }\beta_{aels}\left(k\right)+\alpha_{s}\beta_{SVR}\left(k\right)\right]+\emptyset_{vs}\left(k\right)+\alpha_{1}e_{vs}\left(k\right)\\ \;+\alpha_{2}sgn\;\left(V\left(k\right)\right)\\ \beta_{vs}\left(k\right)={\left(\alpha_{3}\right)}^{T}\gamma_{vs}\left(k\right) \end{array}$$
where $\gamma_{vs}\left(k\right)$ is the state-space of the bearing’s vibration data estimation based on the V-structure technique, $e_{vs}\left(k\right)$ is the error of the bearing’s vibration data estimation based on the V-structure algorithm, $\beta_{vs}\left(k\right)$ is the output of the bearing’s vibration data estimation based on the V-structure approach, $\emptyset_{vs}\left(k\right)$ is the fault estimation part based on the V-structure scheme, V is the V-structure slope using the V-structure approach, and $\left(\alpha_{1},\alpha_{2},\alpha_{3}\right)$ is the coefficient for tuning the V-structure technique. The V-structure slope can be defined using linear techniques such as the proportional-integral (PI) technique, the proportional-derivative (PD) approach, and the proportional-integral-derivative (PID) scheme. In this work, the V-structure approach is defined by the following definition:(14)$$V=\alpha_{4}e_{vs}\left(k\right)+\alpha_{5}\sum e_{vs}\left(k\right)+\alpha_{6}{\dot{e}}_{vs}\left(k\right)$$

Here, $\sum e_{vs}\left(k\right)$ is the integral term of the error of the bearing’s vibration data estimation based on the V-structure algorithm, ${\dot{e}}_{vs}\left(k\right)$ is the derivative term of the error of the bearing’s vibration data estimation based on the V-structure algorithm, and $\left(\alpha_{4},\alpha_{5},\alpha_{6}\right)$ is the coefficient for tuning the PID algorithm. In addition, $e_{vs}\left(k\right)$ and $\emptyset_{vs}\left(k\right)$ can be defined using the following equation:(15)$$\{\begin{array}{l} \emptyset_{vs}\left(k\right)=\alpha_{7}\left(I_{i}\left(k\right)-\beta_{vs}\left(k\right)\right)+\alpha_{2}sgn\;\left(V\left(k\right)\right)+\alpha_{1}sgn\;\left(e_{vs}\left(k\right)\right)\\ e_{vs}\left(k\right)=\beta_{vs}\left(k\right)-\beta_{vs}\left(k-1\right) \end{array}$$
where $\alpha_{7}$ is a coefficient. Moreover, the sign function, (sgn(V)), is introduced by the following definition:(16)$$\{\begin{array}{l} sgn\left(V\right)=1\;if\;V\geq 0\\ sgn\left(V\right)=-1\;if\;V<0 \end{array}.$$

This technique is robust and stable. However, to reduce the issue of oscillation at a high frequency, two methods can be used: (1) using smooth functions instead of the sign function, and (2) using the nonlinear V-structure algorithm instead of the linear PID V-structure approach. In this study, the second method was selected. To improve the performance of the linear PID V-structure function, the nonlinear fuzzy PID V-structure function was recommended. In the classical PID fuzzy algorithm, the fuzzy system has three inputs and one output. If seven conditions are considered for each input, 343 rule bases must be defined for the classical PID fuzzy algorithm. To reduce the number of rule bases in the algorithm, PD fuzzy plus PI fuzzy can be designed instead of PID classical fuzzy. In this method, we have 49 rule bases for PD fuzzy and 49 rule bases for PI fuzzy. Thus, the number of rule-bases in the proposed (PD+PI) fuzzy is 98. Figure 4 illustrates the proposed (PD+PI) fuzzy observer.

The fuzzy V-structure (FVS) technique is defined as the following equation:(17)$$\{\begin{array}{l} \gamma_{fvs}\left(k+1\right)=\left[\alpha_{\gamma }\gamma_{aels}\left(k\right)+\alpha_{i}I_{i}\left(k\right)+\alpha_{\beta }\beta_{aels}\left(k\right)+\alpha_{s}\beta_{SVR}\left(k\right)\right]+\emptyset_{fvs}\left(k\right)+\alpha_{1}e_{fvs}\left(k\right)\\ \;+\alpha_{2}sgn\;\left(V_{f-PID}\left(k\right)\right)\\ \beta_{fvs}\left(k\right)={\left(\alpha_{3}\right)}^{T}\gamma_{fvs}\left(k\right) \end{array}$$

Here, $\gamma_{fvs}\left(k\right)$ is the state-space of the bearing’s vibration data estimation based on the fuzzy V-structure technique, $e_{fvs}\left(k\right)$ is the error of the bearing’s vibration data estimation based on the fuzzy V-structure algorithm, $\beta_{fvs}\left(k\right)$ is the output of the bearing’s vibration data estimation based on the fuzzy V-structure approach, $\emptyset_{fvs}\left(k\right)$ is the fault estimation part based on the fuzzy V-structure scheme, and $V_{f-PID}\left(k\right)$ is the V-structure slope using the PID fuzzy V-structure approach. The fuzzy V-structure slope can be defined based on Figure 4 and the following definition:(18)$$V_{f-PID}\left(k\right)=\alpha_{\mathrm{PD}}V_{f-PD}\left(k\right)+\alpha_{\mathrm{PI}}\sum V_{f-PD}\left(k\right)$$
where $V_{f-PD}\left(k\right)$ is the V-structure slope using the PD fuzzy V-structure approach, and $\left(\alpha_{\mathrm{PD}},\alpha_{\mathrm{PI}}\right)$ are coefficients for the PID fuzzy V-structure. Furthermore, $e_{fvs}\left(k\right)$ and $\emptyset_{fvs}\left(k\right)$ can be defined using the following equation:(19)$$\{\begin{array}{l} \emptyset_{fvs}\left(k\right)=\alpha_{7}\left(I_{i}\left(k\right)-\beta_{fvs}\left(k\right)\right)+\alpha_{2}sgn\;\left(V_{f-PID}\left(k\right)\right)+\alpha_{1}sgn\;\left(e_{fvs}\left(k\right)\right)\\ e_{fvs}\left(k\right)=\beta_{fvs}\left(k\right)-\beta_{fvs}\left(k-1\right) \end{array}$$

To improve the fault estimation in the FVS technique, the fuzzy algorithm is recommended, and the FVSF technique was used. The FVSF estimator can be introduced using the following definition:(20)$$\{\begin{array}{l} \gamma_{fvsf}\left(k+1\right)=\left[\alpha_{\gamma }\gamma_{aels}\left(k\right)+\alpha_{i}I_{i}\left(k\right)+\alpha_{\beta }\beta_{aels}\left(k\right)+\alpha_{s}\beta_{SVR}\left(k\right)\right]+\emptyset_{fvsf}\left(k\right)\\ \;+\alpha_{1}e_{fvsf}\left(k\right)+\alpha_{2}sgn\;\left(V_{f-PID}\left(k\right)\right)\\ \beta_{fvsf}\left(k\right)={\left(\alpha_{3}\right)}^{T}\gamma_{fvsf}\left(k\right) \end{array}$$

Here, $\gamma_{fvsf}\left(k\right)$ is the state-space of the bearing’s vibration data estimation based on the fuzzy V-structure fuzzy fault estimator technique, $e_{fvsf}\left(k\right)$ is the error of the bearing’s vibration data estimation based on the fuzzy V-structure fuzzy fault estimator algorithm, $\beta_{fvsf}\left(k\right)$ is the output of the bearing’s vibration data estimation based on the fuzzy V-structure fuzzy fault estimator approach, and $\emptyset_{fvsf}\left(k\right)$ is the fault estimation part based on the fuzzy V-structure fuzzy fault estimator scheme. Additionally, $e_{fvsf}\left(k\right)$ and $\emptyset_{fvsf}\left(k\right)$ can be defined using the following equation:(21)$$\{\begin{array}{l} \emptyset_{fvsf}\left(k\right)=\alpha_{7}\left(I_{i}\left(k\right)-\beta_{fvsf}\left(k\right)\right)+\alpha_{2}sgn\;\left(V_{f-PID}\left(k\right)\right)+\alpha_{1}sgn\;\left(e_{fvsf}\left(k\right)\right)\\ \;+\alpha_{f}\beta_{f}\left(k\right)\\ e_{fvsf}\left(k\right)=\beta_{fvsf}\left(k\right)-\beta_{fvsf}\left(k-1\right) \end{array}$$
where $\beta_{f}\left(k\right)$ is the output of the bearing’s vibration fault estimation based on the fuzzy algorithm and $\alpha_{f}$ is the fuzzy coefficient. Moreover, if ϵ is introduced by the fuzzy approach is defined as follows.
(22)$$\{\begin{array}{l} if\;e_{fsvf}\left(k\right)\;is\;\left(e_{vsf}\left(k\right)+\epsilon \right)\;then\;\beta_{f}\left(k+1\right)=\beta_{f}\left(k\right)-\alpha_{1}sgn\;\left(e_{fvsf}\left(k\right)\right)\\ if\;e_{fsvf}\left(k\right)\;is\;\left(e_{vsf}\left(k\right)\right)then\;\beta_{f}\left(k+1\right)=\beta_{f}\left(k\right)+\alpha_{x}sgn\;\left(e_{fvsf}\left(k\right)\right)\\ if\;e_{fsvf}\left(k\right)\;is\;\left(e_{vsf}\left(k\right)-\epsilon \right)then\;\beta_{f}\left(k+1\right)=\beta_{f}\left(k\right)+\alpha_{1}sgn\;\left(e_{fvsf}\left(k\right)\right) \end{array}$$

Here, ϵ is the band of error and $\alpha_{x}>\alpha_{1}.$ In the next part, the residual data are generated using the difference between the original bearing’s vibration data and the estimated bearing’s vibration data. The residual generator based on the FVSF technique, $r_{fvsv}\left(k\right),$ is introduced using the following definition:(23)$$r_{fvsv}\left(k\right)=I_{i}\left(k\right)-\beta_{fvsf}\left(k\right)$$

As shown in Figure 1, the vibration’s bearing data were modeled using the combination of AE with the Laguerre filter and SVR technique, (AELS). Next, the hybrid robust estimator was designed to estimate the bearing’s data under unknown conditions. Table 3 outlines the steps for designing the proposed data estimation using the fuzzy V-structure fuzzy fault estimator technique. In the next part, SVM was is used for fault classification and crack size identification.

### 3.3. Data Classification

After residual data generation using the proposed hybrid FVSF estimator, the resampled RMS feature was extracted from the residual data using the following definition:(24)$$r_{fvsv-rms}\left(k\right)=\sqrt{\frac{1}{X}\sum_{j=1}^{X}r_{fvsv}{\left(k\right)}^{2}}$$

Here, $r_{fvsv-rms}\left(k\right)$ and X denote the RMS value for the residual signal based on the proposed FSVF estimator algorithm and the number of windows, respectively. The original data in each condition and state have 120,000 samples. We generated 100 windows for each condition; thus, each window included 1200 samples. SVM was recommended for the RMS residual data classification and crack size identification. It was introduced using the following definition:(25)$$I_{a}\left(\beta^{T}{}_{a}\rho (I_{b})+\beta_{b}\right)\geq I_{a}-D_{i}.$$

Here, $\left(I_{a},\;I_{b}\right)$ is the SVM input for classification and identification, $\left(\beta_{a},\beta_{b}\right)$ is the SVM data classifier/identifier, $\rho (I_{b})$ is the SVM feature, and D is the maximum boundary distance in SVM for data classification and identification. If ρ is introduced as a penalty, to solve the above equation, we have
(26)$$\begin{array}{c} \begin{array}{ll} min & \frac{1}{2} \end{array}\beta^{T}\beta +\rho \underset{j}{\sum }D_{i}\\ \begin{array}{ll} s.t. & \begin{array}{ll} I_{a}\left(\beta^{T}{}_{a}\rho (I_{b})+\beta_{b}\right)\geq I_{a}-D_{i}\; & D_{i}\geq 0 \end{array} \end{array} \end{array}$$

The saddle point is defined using the following definition:(27)$$S_{p}=\frac{1}{2}\beta^{T}\beta +\rho \underset{j}{\sum }D_{i}-\underset{i}{\sum }a_{i}\left[I_{a}\left(\beta^{T}{}_{a}\rho (I_{b})+\beta_{b}\right)-I_{a}+D_{i}\right]-\underset{i}{\sum }b_{i}D_{i}$$
where $S_{p}$ is the min-max value for the saddle point and $\left(a_{i},b_{i}\right)$ is the coefficient to solve the saddle function. Moreover, the quadratic programming can be used to solve the saddle point.
(28)$$\begin{array}{cc} min & \frac{1}{2}\beta^{T}\rho \beta +T^{T}\beta \;\\ s.t. & \begin{array}{l} \underset{i}{\sum }\beta I_{a}=0\\ \begin{array}{ll} 0\leq \beta \leq \rho & \forall i \end{array} \end{array} \end{array}$$
where $T=\left[\begin{array}{c} -1\\ -1\\ \vdots \\ -1 \end{array}\right]$. The SVM data classifier/identifier is represented as follows:(29)$$\beta_{a}=\underset{i}{\sum }I_{a}I_{b}K\left(u_{i},u\right)$$
(30)$$\beta_{b}=\frac{1}{\left|\Delta \right|}\underset{a\in \Delta }{\sum }(I_{a}-\underset{i}{\sum }I_{a}I_{b}K\left(u_{i},u\right))$$
where $K\left(u_{i},u\right)$ and Δ are the kernel function and the support vector, respectively. The support vector can be represented by the following definition:(31)$$\Delta =\left\{i|0\leq D_{i}\leq \rho \right\}$$

The SVM function is represented by the following definition:(32)$$I_{a}=sgn\underset{i}{\sum }I_{a}I_{b}K\left(u_{i},u\right)+D_{i}$$

Table 4 illustrates the training and testing dataset for the proposed FSVF estimator with SVM for the bearing’s data classification and crack size identification.

## 4. Results and Discussion

The proposed hybrid fuzzy V-structure fuzzy approach was used for bearing data classification and crack size identification in this study. To test the power of the proposed hybrid data classification and identification algorithm, this technique, FVSF, was compared with the fuzzy V-structure (FVS) algorithm and the V-structure (VS) algorithm. To test the power of data modeling, the combination of AE with the Laguerre filter and SVR technique (AELS) was compared with AE with the Laguerre filter technique (AEL) and the AE method (AE). Figure 5 illustrates the error of bearing data modeling in a normal condition.

Based on this figure, the error of bearing data modeling for the proposed AELS technique is lower than that of the other two approaches. Thus, that technique was employed for bearing data modeling in this study. In the next step, the bearing data were estimated using the V-structure (VS) approach, fuzzy V-structure (FVS) technique, and proposed hybrid fuzzy V-structure fuzzy (FVSF) technique. Figure 6 shows the residual bearing data for the VS technique.

As depicted in the figure, the VS technique has a problem classifying the inner and outer data. To improve the performance of the VS technique, the FVS technique was leveraged, as illustrated in Figure 7.

Figure 8 demonstrates the bearing’s residual signal using the FVSF algorithm. Based on the comparison between Figure 6, Figure 7 and Figure 8, the data classification accuracy using the proposed FVSF algorithm is better than that of the other two approaches. The RMS bearing’s residual signals for the VS technique, the FVS method, and the proposed FVSF approaches are illustrated in Figure 9, Figure 10 and Figure 11, respectively. Based on Figure 9, the VS has a critical issue in classifying the inner and outer faults. The FVS technique (Figure 8) improves the classification accuracy compared with the VS algorithm; however, its limitations in the classification of inner and outer faults remain. Based on these figures, the power of discrimination provided by the proposed FVSF algorithm (Figure 11) is much better than that of the other two methods, and the normal and abnormal signals can be easily classified.

Figure 12, Figure 13 and Figure 14 present confusion matrices of bearing data classification using the combination of VS and SVM, that of FVS and SVM, and that of the proposed FVSF and SVM. Based on Figure 12 and Figure 13, the main misclassification parts for bearing data classification using the combination of VS and SVM and that of FVS and SVM are between inner and outer faults. According to Figure 14, the combination of FVSF and SVM resolves the problem of misclassification.

Furthermore, the average accuracy of the bearing’s data classification based on the combination of VS and SVM, the combination of FVS and SVM, and the combination of the proposed FVSF and SVM is represented in Table 5. According to these figures and Table 4, when the crack sizes are 0.007 in, 0.014 in, and 0.021 in, and the motor torque loads are 0 hp, 1 hp, 2 hp, and 3 hp, the proposed FVSF approach improves the accuracy of the VS technique and FVS method by 10.75% and 5.5%, respectively. Thus, the performance of bearing data classification using the combination of the proposed FVSF and SVM is much better than that of the other two techniques.

Table 6 shows the performance of the bearing’s data identification using the combination of VS and SVM, the combination of FVS and SVM, and the combination of the proposed FVSF and SVM. According to this table, when the motor torque loads are changed between 0 hp and 3 hp, the proposed FVSF approach improves the accuracy of the VS technique and FVS method by 11.55% and 5.2%, respectively. Hence, the performance of the bearing’s crack size identification using the combination of the proposed FVSF and SVM is much better than that of the other two approaches.

In the next part, to validate the effectiveness of the proposed FVSF algorithm, this approach is compared with the following three existing methods: smooth sliding mode digital twin (SSDT) [29], strict feedback backstepping digital twin (SBDT) [35], and multivariable fuzzy learning backstepping (MFLB) [36].

In ref. [29], the authors have used the combination of autoregressive with a Laguerre filter, intelligent gaussian regression, and an intelligent smooth sliding observer for bearing fault diagnosis. Apart from the accuracy and reliability of the proposed method, the smooth algorithm uses a saturation function to reduce the effect of high-frequency oscillations; this algorithm has a lack of robustness. In addition, in ref. [35], the combination of support vector regression, Gaussian Process Regression, and an intelligent integral strict feedback backstepping observer was suggested for fault diagnosis in the bearing. The main challenge in this approach was robustness compare with robust algorithm such as V-structure approach. Furthermore, a multivariable fuzzy learning backstepping observer has been used for fault diagnosis in ref. [36]. However, the nonlinear autoregression technique was used for system modeling, but this technique suffers from low accuracy in crack size detection. To validate the proposed approach further, we calculate the average diagnostic accuracy for the proposed FVSF, SSDT [29], SBDT [35], and MFLB [36] under various operating conditions (see Table 7). Table 7 presents the diagnostic accuracy of the proposed FVSF, SSDT [29], SBDT [35], and MFLB [36] for fault diagnosis in bearing. The diagnostic accuracy is reported as the percentage of correct detection in all data.

As shown in Table 7, the proposed FVSF fault diagnosis method outperforms the state-of-the-art SSDT method, SSDT technique, and SBDT approach, yielding average performance improvements of 1.5%, 1.25%, and 1.75%, respectively. This performance improvement can be further validated by the fact that our proposed FVSF scheme is highly sufficient to identify the fault in the bearing.

## 5. Conclusions

In this study, the hybrid fuzzy V-structure fuzzy fault estimator algorithm was leveraged for bearing anomaly diagnosis. The design of the proposed hybrid technique consisted of three main steps: data modeling, data estimation, and data classification. First, the bearing’s data in normal conditions were modeled using the combination of autoregression with error feedback, the Laguerre filter to improve robustness, and the SVR algorithm to improve the accuracy of nonlinear signal modeling. Next, the bearing’s data were estimated using the proposed hybrid fuzzy V-structure fuzzy fault estimator algorithm. The V-structure observer was selected because of its reliability, robustness, and stability. The fuzzy technique was employed to (1) reduce the high-frequency oscillation and improve robustness and (2) reduce the error of fault estimation and increase the rate of discrimination in different classes. Finally, SVM was used to classify the RMS residual data, which are the difference between the RMS original data and estimated ones. The Case Western Reserve University bearing’s dataset was used to test the power of the proposed hybrid technique. The proposed hybrid fuzzy V-structure fuzzy fault estimator algorithm for fault diagnosis and crack size identification was compared with the fuzzy V-structure approach and the V-structure method. The proposed hybrid approach improved the performance of the bearing’s data fault classification by 5.5% and 10.75% compared with the fuzzy V-structure technique and the V-structure method, respectively. In addition, for crack size identification, the proposed hybrid method improved the performance of the fuzzy V-structure technique and the V-structure method by 5.2% and 11.55%, respectively. In future work, the presented parallel hybrid estimation technique will be enhanced for multi-crack fault diagnosis.

## Figures and Tables

**Figure 1 sensors-23-01021-f001:**
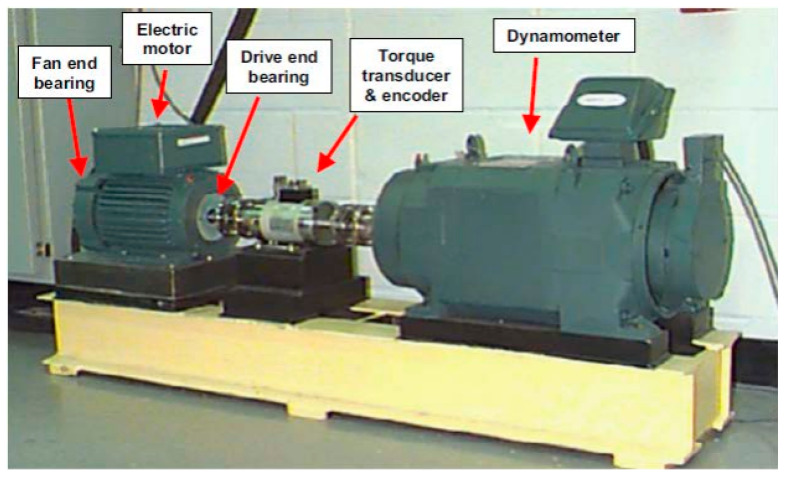
Case Western Reserve University benchmark for recording bearing dataset.

**Figure 2 sensors-23-01021-f002:**
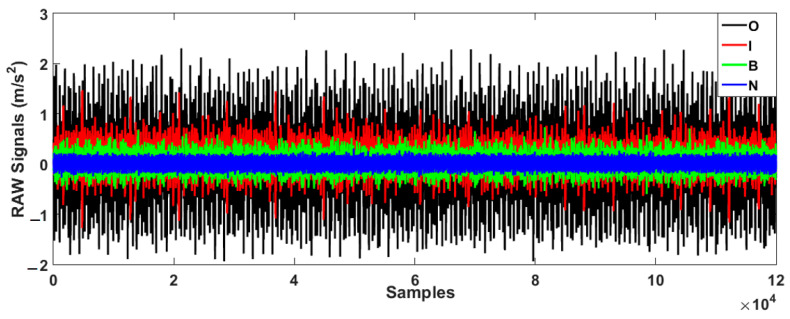
Raw vibration bearing signals in normal and abnormal conditions.

**Figure 3 sensors-23-01021-f003:**
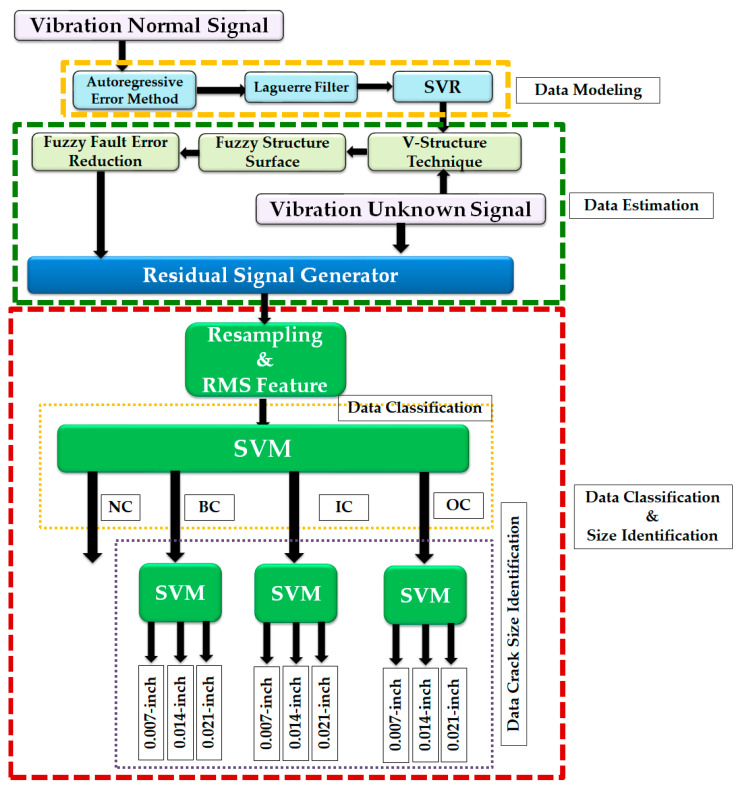
Bearing data classification and data crack size identification using the proposed improved fuzzy V-structure algorithm.

**Figure 4 sensors-23-01021-f004:**
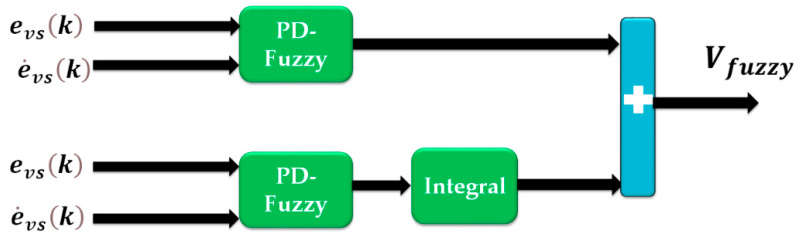
Proposed PID fuzzy observer to evaluate the PID V-structure.

**Figure 5 sensors-23-01021-f005:**
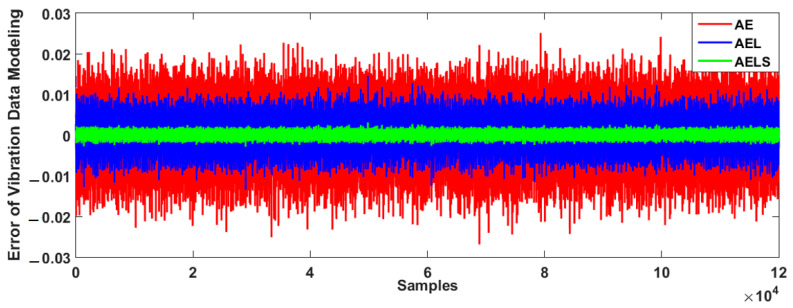
Error of vibration data modeling using AE, AEL, and proposed AELS techniques.

**Figure 6 sensors-23-01021-f006:**
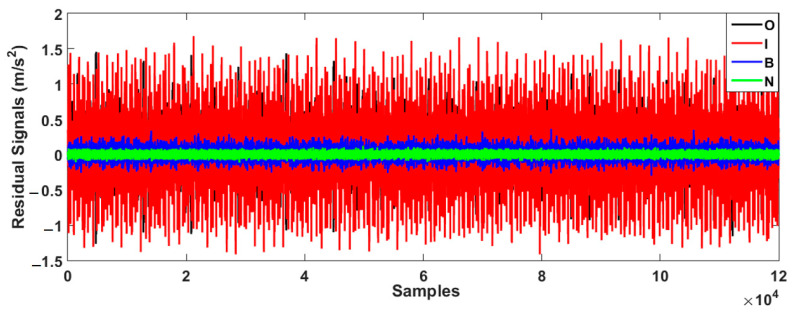
Bearing residual signals using the VS approach.

**Figure 7 sensors-23-01021-f007:**
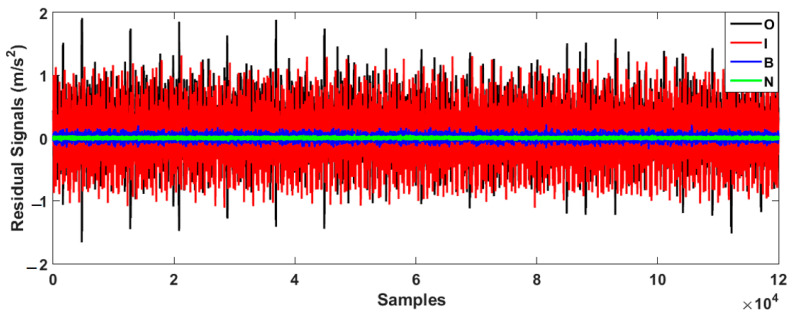
Bearing residual signals using the FVS approach.

**Figure 8 sensors-23-01021-f008:**
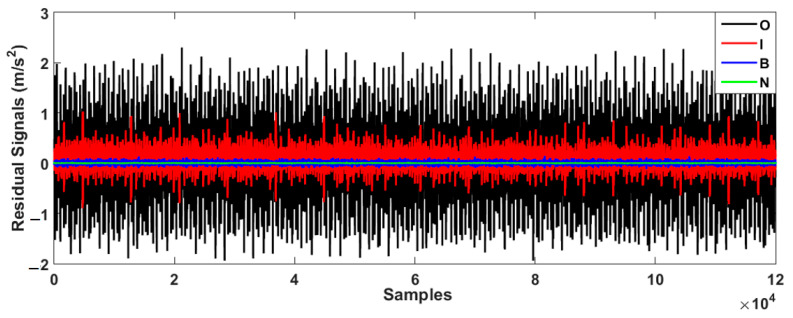
Bearing’s residual signals using the FVSF approach.

**Figure 9 sensors-23-01021-f009:**
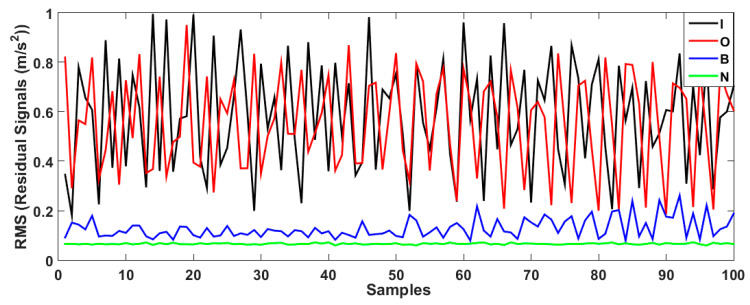
Bearing’s RMS residual signals using the VS technique.

**Figure 10 sensors-23-01021-f010:**
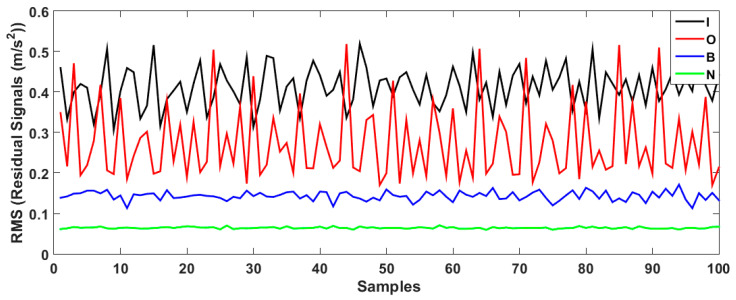
Bearing’s RMS residual signals using the FVS method.

**Figure 11 sensors-23-01021-f011:**
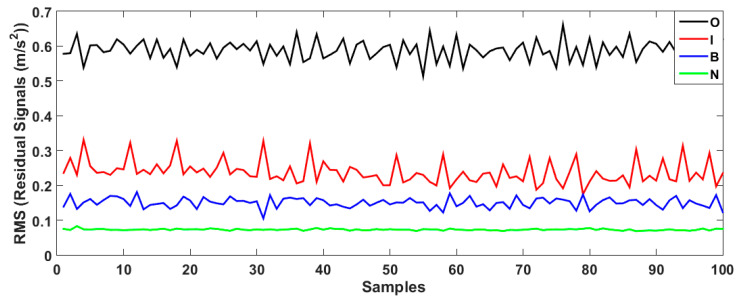
Bearing’s RMS residual signals using the proposed FVSF method.

**Figure 12 sensors-23-01021-f012:**
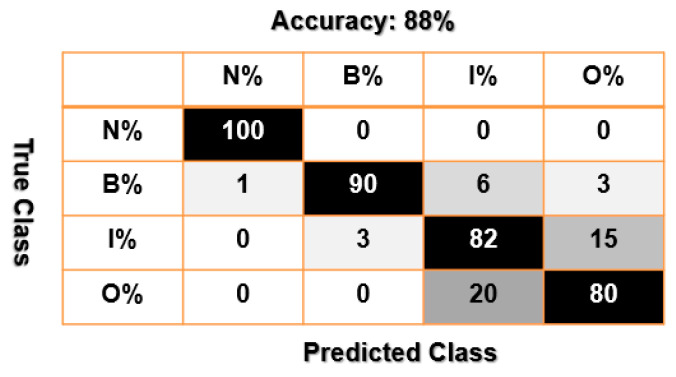
Bearing’s data classification using the combination of VS and SVM.

**Figure 13 sensors-23-01021-f013:**
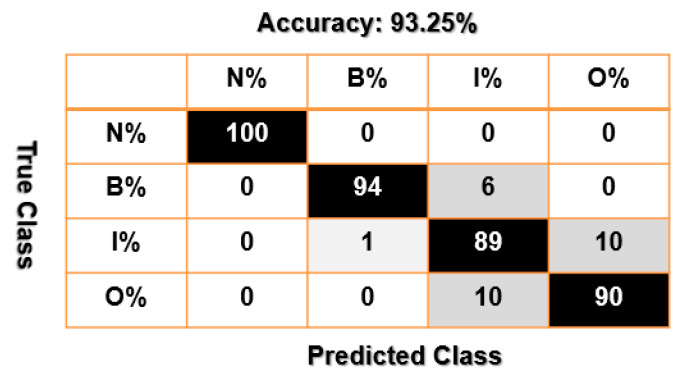
Bearing’s data classification using the combination of FVS and SVM.

**Figure 14 sensors-23-01021-f014:**
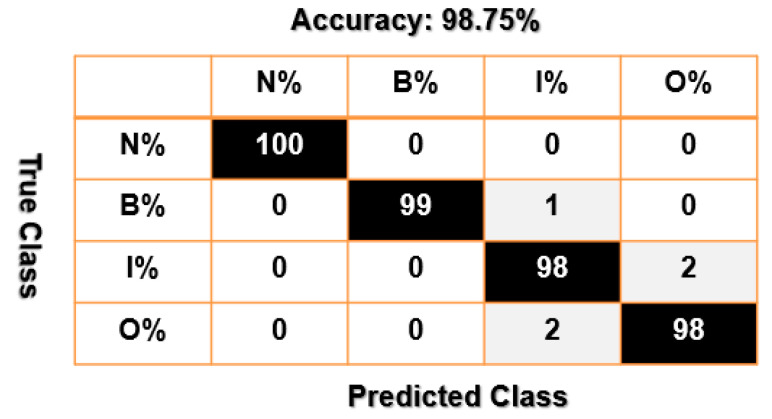
Bearing’s data classification using the combination of the proposed FVSF and SVM.

**Table 1 sensors-23-01021-t001:** Bearing data information for four conditions.

Conditions	Load [hp]	Size of Crack [inches]
N	0, 1, 2, 3	-
B	0, 1, 2, 3	0.007, 0.014, 0.021
I	0, 1, 2, 3	0.007, 0.014, 0.021
O	0, 1, 2, 3	0.007, 0.014, 0.021

**Table 2 sensors-23-01021-t002:** Proposed data modeling using the combination of AE with the Laguerre filter and SVR technique.

1:	Data modeling using the autoregressive procedure; Equations (1) and (2)
	**Detail**
1.1	Calculate $\emptyset_{a}\left(k\right)\leftarrow I_{i}\left(k\right)-\beta_{ae}\left(k\right)$, Equation (2)
1.2	Calculate ${\;\gamma }_{a}\left(k+1\right)\leftarrow \left[\alpha_{\gamma }\gamma_{a}\left(k\right)+\alpha_{i}I_{i}\left(k\right)\right]+\emptyset_{a}\left(k\right)$, Equation (1)
1.3	Calculate $\beta_{a}\left(k\right)\leftarrow {\left(\alpha_{o}\right)}^{T}\gamma_{a}\left(k\right)$, Equation (1)
2:	Modifying the AE; Equations (3) and (4)
	**Detail**
2.1	Calculate $e_{ae}\left(k\right)\leftarrow \beta_{ae}\left(k\right)-\beta_{ae}\left(k-1\right)$, Equation (4)
2.2	Calculate $\emptyset_{ae}\left(k\right)\leftarrow I_{i}\left(k\right)-\beta_{ae}\left(k\right)$, Equation (4)
2.3	Calculate $\gamma_{ae}\left(k+1\right)\leftarrow \left[\alpha_{\gamma }\gamma_{ae}\left(k\right)+\alpha_{i}I_{i}\left(k\right)\right]+\emptyset_{ae}\left(k\right)+\alpha_{e}e_{ae}\left(k\right)$, Equation (3)
2.4	Calculate $\beta_{ae}\left(k\right)\leftarrow {\left(\alpha_{o}\right)}^{T}\gamma_{ae}\left(k\right)$, Equation (3)
3:	Improve the robustness: AE with Laguerre filter (AEL) technique; Equations (5) and (6)
	**Detail**
3.1	Compute $e_{ael}\left(k\right)\leftarrow \beta_{ael}\left(k\right)-\beta_{ael}\left(k-1\right)$, Equation (6)
3.2	Calculate $\emptyset_{ael}\left(k\right)\leftarrow I_{i}\left(k\right)-\beta_{ael}\left(k\right)$, Equation (6)
3.3	Compute $\gamma_{ael}\left(k+1\right)\leftarrow \left[\alpha_{\gamma }\gamma_{ael}\left(k\right)+\alpha_{i}I_{i}\left(k\right)+\alpha_{\beta }\beta_{ael}\left(k\right)\right]+\emptyset_{ael}\left(k\right)+\alpha_{e}e_{ael}\left(k\right)$, Equation (5)
3.4	Compute $\beta_{ael}\left(k\right)\leftarrow {\left(\alpha_{o}\right)}^{T}\gamma_{ael}\left(k\right)$, Equation (5)
4:	Improve the nonlinear system modeling: combination of AE with Laguerre filter and SVR technique; Equations (7) and (8)
	**Detail**
4.1	Compute $e_{aels}\left(k\right)\leftarrow \beta_{aels}\left(k\right)-\beta_{aels}\left(k-1\right)$, Equation (8)
4.2	$\emptyset_{aels}\left(k\right)\leftarrow I_{i}\left(k\right)-\beta_{aels}\left(k\right),$ Equation (8)
4.3	Compute $\gamma_{aels}\left(k+1\right)\leftarrow \left[\alpha_{\gamma }\gamma_{aels}\left(k\right)+\alpha_{i}I_{i}\left(k\right)+\alpha_{\beta }\beta_{aels}\left(k\right)\right]+\emptyset_{ael}\left(k\right)+\alpha_{e}e_{aels}\left(k\right)+\alpha_{s}\beta_{SVR}\left(k\right),$ Equation (7)
4.4	Compute $\beta_{aels}\left(k\right)\leftarrow {\left(\alpha_{o}\right)}^{T}\gamma_{aels}\left(k\right),$ Equation (7)

**Table 3 sensors-23-01021-t003:** Proposed hybrid data estimation using the fuzzy V-structure fuzzy fault estimator.

1:	Data estimation using the V-structure algorithm; Equations (13)–(15)
	**Detail**
1.1	Calculate $e_{vs}\left(k\right)\leftarrow \beta_{vs}\left(k\right)-\beta_{vs}\left(k-1\right)$, Equation (15)
1.2	Calculate $\;\emptyset_{vs}\left(k\right)\leftarrow \alpha_{7}\left(I_{i}\left(k\right)-\beta_{vs}\left(k\right)\right)+\alpha_{2}sgn\;\left(V\left(k\right)\right)+\alpha_{1}sgn\;\left(e_{vs}\left(k\right)\right)$, Equation (15)
1.3	Calculate $V\leftarrow \alpha_{4}e_{vs}\left(k\right)+\alpha_{5}\sum e_{vs}\left(k\right)+\alpha_{6}{\dot{e}}_{vs}\left(k\right)$, Equation (14)
1.4	Calculate $\gamma_{vs}\left(k+1\right)\leftarrow \left[\alpha_{\gamma }\gamma_{aels}\left(k\right)+\alpha_{i}I_{i}\left(k\right)+\alpha_{\beta }\beta_{aels}\left(k\right)+\alpha_{s}\beta_{SVR}\left(k\right)\right]+\emptyset_{vs}\left(k\right)+\alpha_{1}e_{vs}\left(k\right)$+$\alpha_{2}sgn\;\left(V\left(k\right)\right)$, Equation (13)
1.5	Calculate $\beta_{vs}\left(k\right)\leftarrow {\left(\alpha_{3}\right)}^{T}\gamma_{vs}\left(k\right),$ Equation (13)
2:	Reduce high frequency oscillation: PID fuzzy V-structure algorithm; Equations (17)–(19)
	**Detail**
2.1	Compute $e_{fvs}\left(k\right)\leftarrow \beta_{fvs}\left(k\right)-\beta_{fvs}\left(k-1\right)$, Equation (19)
2.2	Calculate $\emptyset_{fvs}\left(k\right)\leftarrow \alpha_{7}\left(I_{i}\left(k\right)-\beta_{fvs}\left(k\right)\right)+\alpha_{2}sgn\;\left(V_{f-PID}\left(k\right)\right)+\alpha_{1}sgn\;\left(e_{fvs}\left(k\right)\right)$, Equation (19)
2.3	Compute $V_{f-PID}\left(k\right)\leftarrow \alpha_{\mathrm{PD}}V_{f-PD}\left(k\right)+\alpha_{\mathrm{PI}}\sum V_{f-PD}\left(k\right)$, Equation (18)
2.4	Compute $\gamma_{fvs}\left(k+1\right)\leftarrow \left[\alpha_{\gamma }\gamma_{aels}\left(k\right)+\alpha_{i}I_{i}\left(k\right)+\alpha_{\beta }\beta_{aels}\left(k\right)+\alpha_{s}\beta_{SVR}\left(k\right)\right]+\emptyset_{fvs}\left(k\right)+\alpha_{1}e_{fvs}\left(k\right)+\alpha_{2}sgn\;\left(V_{f-PID}\left(k\right)\right)$, Equation (17)
2.5	Compute $\beta_{fvs}\left(k\right)\leftarrow {\left(\alpha_{3}\right)}^{T}\gamma_{fvs}\left(k\right)$, Equation (17)
3:	Improve the fault estimation: PID fuzzy V-structure fuzzy algorithm; Equations (20)–(22)
	**Detail**
3.1	Compute $e_{fvsf}\left(k\right)=\beta_{fvsf}\left(k\right)-\beta_{fvsf}\left(k-1\right),$ Equation (21)
3.2	Compute $\emptyset_{fvsf}\left(k\right)=\alpha_{7}\left(I_{i}\left(k\right)-\beta_{fvsf}\left(k\right)\right)+\alpha_{2}sgn\;\left(V_{f-PID}\left(k\right)\right)+\alpha_{1}sgn\;\left(e_{fvsf}\left(k\right)\right)+\alpha_{f}\beta_{f}\left(k\right),$ Equation (21)
3.3	Compute $\gamma_{fvsf}\left(k+1\right)=\left[\alpha_{\gamma }\gamma_{aels}\left(k\right)+\alpha_{i}I_{i}\left(k\right)+\alpha_{\beta }\beta_{aels}\left(k\right)+\alpha_{s}\beta_{SVR}\left(k\right)\right]+\emptyset_{fvsf}\left(k\right)+\alpha_{1}e_{fvsf}\left(k\right)+\alpha_{2}sgn\;\left(V_{f-PID}\left(k\right)\right),$ Equation (20)
3.4	Compute $\beta_{fvsf}\left(k\right)\leftarrow {\left(\alpha_{3}\right)}^{T}\gamma_{fvsf}\left(k\right),$ Equation (20)

**Table 4 sensors-23-01021-t004:** Training and testing RMS residual data.

			Training Samples	Testing Samples
Data Classification		N	300	100
		B	900	300
		I	900	300
		O	900	300
Data crack size identification	B	0.007 inch	300	100
		0.014 inch	300	100
		0.021 inch	300	100
	I	0.007 inch	300	100
		0.014 inch	300	100
		0.021 inch	300	100
	O	0.007 inch	300	100
		0.014 inch	300	100
		0.021 inch	300	100

**Table 5 sensors-23-01021-t005:** Comparison of the combination of VS and SVM, the combination of FVS and SVM, and the combination of the proposed FVSF and SVM for bearing data classification.

States	Combination of VS and SVM (%)	Combination of FVS and SVM (%)	Combination of FVSF and SVM (%)
N	100	100	100
B	90	94	99
I	82	89	98
O	80	90	98
Average classification accuracy	88	93.25	98.75

**Table 6 sensors-23-01021-t006:** Comparison of the combination of VS and SVM, the combination of FVS and SVM, and the combination of the proposed FVSF and SVM for bearing crack size identification.

State	Crack Sizes-Inch	Combination of VS and SVM (%)	Combination of FVS and SVM (%)	Combination of FVSF and SVM (%)
B	0.007	84	90	97
	0.014	86	92	98
	0.021	85	92	98
I	0.007	82	93	98
	0.014	88	94	98
	0.021	88	92	98
O	0.007	89	94	98
	0.014	87	93	99
	0.021	89	95	98
Average accuracy of bearing’s data crack size identification		86.45	92.8	98

**Table 7 sensors-23-01021-t007:** Comparison of the proposed FVSF, SSDT [29], SBDT [35], and MFLB [36] for bearing data classification.

States	FVSF (%)	SSDT [28] (%)	SBDT [34] (%)	MFLB [35] (%)
N	100	100	100	100
B	99	94	96	95
I	98	98	96	98
O	98	97	98	95
Average classification accuracy	98.75	97.25	97.5	97

## Data Availability

Public dataset.

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
