# Peer review of "Bearing Fault Diagnosis Using a Hybrid Fuzzy V-Structure Fault Estimator Scheme"

_sensors, 2023, doi:10.3390/s23021021_

Round 1

Reviewer 1 Report

This paper discussed a bearing fault diagnosis method called the hybrid fuzzy V-structure fault estimator scheme. The method is the combination of a lot of existing methods, including the fuzzy algorithm, autoregression, Laguerre filter, support vector regression, support vector machine. Generally, the organization is very poor and the English writing is messy. 

(1)   Abstract: Line 11, two fuzzy words are used.

Line 13-15: reduce the oscillation…. The meaning is unclear. What oscillation is reduced?

The whole abstract part should be re-organized to emphasize the key work.

(2)   Line 39: authors listed four groups of methods. However, authors do not give the detailed explanations. What is the difference between the signal-based and AI-based methods? Don’t the AI-based methods apply the signals?

(3)   Line 47: a lot of AI-based methods have been proposed. However, authors only mentioned the fuzzy logic and neural network.

(4)   Authors designed a method in this paper, What is the motivation? What problems do author handle? Authors listed three contributions in line 113-118. The used techniques are the existing ones. What are the novelty points?

(5)   In Eq. (1), what is the output \beta_a(k)? How do authors determine the model parameters?

(6)   What is the V-structure estimator? Authors do not explain this clearly. Even no related references are cited.

(7)   The used testing dataset is a well-known open dataset. Authors should compare the results with the other existing state-of-the-art methods.

Reviewer 2 Report

1.     The innovation of this paper should be highlighted in Introduction. Compared with the existing methods, what is the main advantage of the proposed method?

2.     Some studies are discussed and your method is proposed in Section Introduction. But what is the logical relationship between them? What are the problems in the existing research? Does the proposed method solve the existing problems?

3.     It should be explained in detail the purposes of data modeling and data estimation in the proposed method.

4.     The comparison should be conducted between the proposed method and several benchmark methods or several advanced methods, rather than some simple algorithms. It is obvious that the VS and FVS algorithms are simpler than the proposed FVSF algorithm. So the comparison is not fair.

Reviewer 3 Report

I found your article very interesting titled “Bearing Fault Diagnosis Using a Hybrid Fuzzy V-Structure Fault Estimator Scheme”, but in my opinion below remarks would improve your manuscript under the scientific level.

Comments and Suggestions for Authors:

1.       In the Abstract I suggest to focus only on what was done briefly and on novelties. In my opinion, the Abstract is a little bit too long.

2.       I don’t agree with the fact in the Introduction, that most rolling-element bearing’s research is based on the constant operating conditions. I suggest to refer to the 2 following references, which I recommend to cite:

·       A new statistical features based approach for bearing fault diagnosis using vibration signals. Sensors, 2022, 22(5), 2012.

·       The influence of the radial internal clearance on the dynamic response of self-aligning ball bearings. Mechanical Systems and Signal Processing, 2022, 171, 108954.

3.       Regarding the experimental setup, please put into the table form. In the present form, all information is distributed over the whole paragraph. In my opinion, it will improve the presentation of the test.

4.       Moreover, please add the information about the data acquisition. Type of accelerometers and sampling time.

5.       The results are quite impressive, but what can be the reason of difference in classification between faults of balls and outer ring?

6.       In my opinion, the linear analysis with Fourier transform is missing, that there is no information on characteristic frequencies corresponding to specific fault in bearing.

7.       Without doubt, the V-structure fuzzy fault estimator is quite reliable tool for the fault estimation in bearings. However, I would like to ask about your planned future studies for the analysis of multi-crack fault diagnosis. Are you going to use your test rig? If yes, I would add this information for your project reason to disseminate it.

Round 2

Reviewer 1 Report

Authors have improved the paper clearly.  I think this paper can be accepted after solving the following small problems:

(1)The abbreviation words should be defined when they are used for the first time. E.g. PID in line125. Furthermore, it is not necessary to define them repeatedly. 

(2) The text style in Figs. 12-14 should be consistent with the used in other figures.
